# Preaching as Protest against the Apophatic Silencing of God's People

**Will Willimon**

Duke Divinity School, Duke University, Durham, NC 27708, USA; will@duke.edu

**Abstract:** Throughout church history, there have been those who stressed the limits of our ability to speak with confidence about God and extolled the nobility of silence in the face of God's ineffability. Dionysius the Areopagite famously asserted, "With regard to the divine, negations are true, whereas affirmations are inadequate". Apophatic silence is presented as respectful of the mysterious otherness of God. Christian preaching is a practice that refutes all attempts at negative, apophatic theology. Every sermon participates in the wonder of the uniquely Jewish and Christian claim that God not only speaks but also invites, even commands, humanity to speak about God as well. Christian preaching is suspicious of any attempt to sentimentalize silence in the name of humble acknowledgement of human limitations to speak truthfully about God. Preaching therefore requires the courage to speak up and speak out with the God who, in Jesus Christ, has spoken to us. The silencing of the voices of women, persons of color, and others who claim to know that God is with them is an aspect of neocolonial oppression that preaching cannot abide. Preaching is a protest against all those who would tell the voiceless that some things are better left unsaid.

**Keywords:** preaching; homiletics; apophatic; negative theology; Christian; Chalcedon; Karl Barth; the Trinity

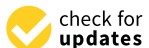



## 1. Introduction: Christian Preaching as Public Speech

Christian public speech, otherwise known as preaching, has always been at the heart of the Christian faith. Preaching is undertaken in the conviction that it is the nature of the God whom Christians and Jews worship to self-present through communication. God is Triune—Father, Son, and Holy Spirit. The three persons of the Trinity are eternally in conversation with one another and with creation. The God of Israel and the church creates and continues to sustain the cosmos through speech. The beginning of everything? *Deus dixit*, "God said . . ." (Gen 1:3).

Thus, Jesus came preaching (Mark 1:14). The gospel (*euanggelion*) is good news, public proclamation of the significance of Jesus Christ first by Christ and then by those who have received news of Christ. As Søren Kierkegaard said, "Christianity does not arise from any human heart" (Kierkegaard 1985, p. 109). Nor can the Creator be accessed through creation's nonverbal silence. The gospel is news, a *verbum externum*, an "external word" (Anonymized, p. 28), a word that is spoken to us rather than arises out of us. One comes to the truth about God through audition and verbal reception. A Christian is someone who has heard and believed the truth about God, a gift, news that can be received only by the truth having been told to you. Little of this faith is self-derived. Thus, Lamin Sanneh has characterized Christianity as training in the art of receiving news of God from another (Sanneh 2003).

The news that is preached about Christ is a public pronouncement that demands to be announced to others. Jesus said to his disciples, "Don't be afraid of those people" (Matt 10:26; all scripture quotes are from the Common English Bible). Afraid of what? Afraid to "announce", that is, to preach. "Nothing is hidden that won't be revealed, and nothing secret that won't be brought out into the open. What I say to you in the darkness, tell in the

light; and what you hear whispered, announce from the rooftops" (Matt 10:26–27). What is said secretly, privately, individually, is to be publicly announced "from the rooftops".

"Don't be afraid of those people". Which people? Did Jesus command his followers to engage in public speech (preaching) because some were tempted to keep the news they had heard to themselves as if it were secret gnosis? Or was it fear of censure from opponents that kept them from speaking up and speaking out? Many early preachers would pay a high price for their bold speech about Jesus.

"For Zion's sake I won't keep silent" (Isa 62:1): this ancient prophetic commitment to bear witness to YHWH's faithfulness to Israel holds, too, for the Christian preacher who yearns to speak of the Trinitarian God.

## 2. The Lure of the Apophatic

Although preaching—making public claims about God—is inherent in the Christian faith, as early as the fifth century, there were those who claimed that it is an offense against the nature of God to make verbal, positive, public assertions about God's nature and work. Pseudo-Dionysius the Areopagite, a neoplatonic theologian of the late fifth to the early sixth century, famously asserted, "With regard to the divine, negations are true, whereas affirmations are inadequate" (Dionysius the Areopagite 1970, p. 79). Mortals can say for sure only what God is not, not what God is. In this, Pseudo-Dionysius continued a line of reasoning rooted in Plato, then Philo and Plotinus.

Dionysius asserted the inexpressibility of God, not on the basis of the limits of human language linguistically to ascend toward the divine, but rather because of the nature of God. God is a priori defined as beyond the reach of human verbalization, beyond definition, remote, nonhuman, incorporeal and therefore inarticulate in ways that can be comprehended by humans. The divine can be accessed only through nonverbal contemplation and by disposing of all less-than-adequate verbal definitions of the divine.

Multiple theologians down through the ages have warned of the idolatrous dangers of thoughtless anthropomorphism in our discourse about God. But apophatics go beyond that, impugning the possibility of finding accurate words about God, or at least words about a God as Jews, Christians, and Muslims have spoken of God.

Pseudo-Dionysius became influential in the contemplative tradition of the eastern Orthodox Churches and had a significant impact on western mysticism. Statements about God can only say negatively what God is not (apophatic), rather than positively account for what God is (cataphatic). Thus began the apophatic tradition in Christianity.

Dionysius' concept of God as unknowable, unrestricted beyond individual substances, beyond even our highest concepts of goodness, draws upon Neoplatonism. Human ascent toward divinity is a process of negation, purgation of our inadequate words about God. Apophatism is a product of Greek ways of thinking, which Dionysius found in Neoplatonism. The way to think is through detached contemplation of the ideal. Cataphatic theology contends that the way to know something is through relationship, encountering something in all its otherness, with the humble acknowledgement that there is something there that is not constructed by the self.

The German Dominican Meister Eckhart (ca. 1260–ca. 1328) extolled silence as next to godliness when he declared that there is so great a chasm between God and humanity that "nothing creaturely is so like God as silence" (Eckhart 1955, p. 367). He lectured preachers, "be silent and do not chatter about God; for when you chatter . . . you are telling lies and sinning" (Eckhart 1981, p. 207). Eckhart characterized as "brutish" any attempt to "understand God who is beyond words" (Eckhart 1981, pp. 206–7).

Apophatics like Pseudo-Dionysius and Meister Eckhart asserted that we can only say for sure what God is not because God is, by definition, incapable of definition (Deus definiri nequit).

Mainstream Christian theologians, particularly Protestant theologians, rejected apophatism. Martin Luther preferred Deus absconditus (suggested by Isa 45:15) to Dionysius' Deus incognitus. Luther dismissed Eckhart's "wholly unknown" God for Christological reasons.

God is elusive to us, not because of God's ontological distance from humanity, but rather because the God who comes to us in the crucified Christ is inconceivable to humans due to our misconceptions of God. Whoever the God was that Dionysius said cannot be known, this God has nothing to do with the verbally revealing Christ, said Luther (Rorem 1997).

## 3. Contemporary Apophatics

While the apophatic approach to God has never supplanted the dominance of the cataphatic in Western Christianity, there has been a curious resurgence of negative theology in the last couple of decades. Karen Armstrong is an example of a modern successor to the apophatic tradition. In her book The Case for God, Armstrong praises the recovery of apophatic theology in "postmodern theology" (Armstrong 2009). Positive statements about God should be made only with great caution, humbly recognizing our human limitations to say something definitive about God. Apophatic humility is the best path to conversation between different faiths. Like the apophatics before her, Armstrong says that God is encountered in silence on the far side of language and knowledge. Unlike classical apophatics, Armstrong stresses that "postmodern" access to God is not so much through contemplation and meditation as through nonverbal "Christian practices", including contemplation and meditation.

Philosopher and literary scholar William Franke lauds negative theology's accentuation of language's limitations as being not simply one aspect of the history of religious thought, but as the "perennial counter-philosophy to the dominant [and, Franke thinks misguided] philosophy of Logos" whose verbosity holds Western religious thought in thrall (Franke 2020, p. 1). He wants language to be "infinitely open" to revelation. There is no way for theology to avoid premature and debilitating closure without negative theology's cleansing of baseless definitions of the divine. The irony is that Franke's "infinite openness" does not include openness to the possibility that we can openly speak accurately about God. He also has great faith in his mind's ability to purge itself of preconceptions and thereby achieve "infinite openness".

Silence is "praised by all religions", claims Dale Allison. "Maybe we have murdered God . . . finally did away with [God] indirectly, by exterminating silence. Artificial noise has become an unholy liturgy that unites all . . ., [drawing us away] from nature's God and [God's] self-imposed muteness of love" (Allison 1995, pp. 36, 37, 40). Nature is silent? Love is mute?

In her 1997 Beecher lectures, Barbara Brown Taylor warned preachers that many in Christian congregations "have become suspicious if not downright disdainful of the spoken word" (Taylor 1998, p. 33). Deluged by a sea of words (due mostly to developments in communication technology), our words have become cheap and have lost their meaning. Taylor charges that our talk about God is particularly suspect because of "God's own silence. If God spoke directly to people, preachers could retire. As it is, God's reticence is the problem the clergy think they are hired to address" (Taylor 1998, p. xi).

Taylor judges that the silence of God is the major pastoral challenge. When we cry out to God because Scripture tells us that, "the faithful receive what they ask for: children, manna, land, health. By implication, those who do not receive are not faithful. . . If they were, God would speak to them. 'For everyone who asks receives, and everyone who searches finds, and for everyone who knocks, the door will be opened'" (Luke 11:10). "This is the condemnation that hangs over the silence of God . . ." (Taylor 1998, p. 52). Taylor complains that contemporary preachers have flooded the world with a barrage of compensatory words, so intimidated are we by the frequent silence of God.

Philosopher Ludwig Wittgenstein's statement, "Whereof one cannot speak, therefore one must be silent", is often used as justification for silence when it comes to statements about God (Wittgenstein 2001, p. 89). God is defined as the one whose greatness renders all talk of God inadequate, beyond all human words: God, the ineffable mystery beyond verbalization.

Robert Sarah has much to say about the holiness of saying nothing, calling the modern world an oppressive "dictatorship of noise" and "symptom of serious, worrisome illness" (Sarah 2016, p. 27). "Solitude and silence . . . is where God dwells. [God] drapes himself in silence" (Sarah 2016, p. 30). In speaking up for silence, Sarah is more indebted to Friedrich Schleiermacher than traditional Catholic teaching. Schleiermacher sought to distinguish true religion from fake by making the mark of true religion sincerity, earnest interiority. Religion is to be judged by the intensity of the individual's personal experience; religion safely tucked into the realm of the tamed human heart. True religion, said Schleiermacher, "is only seen in secret by those who love it". True religion is invisible and ultimately ineffable.

Times of silence in the liturgy ought to foster receptivity to God's speaking. Yet in churches in North American culture, malformed listeners are likely to use liturgical silence as another occasion to fill vacuous space with individual yearnings. Sin is apt to infect human silence no less than human speech. Moments of silence can be little more than encouragement for worshippers to dive deeper into self-absorption. Lack of speech, verbalization does not ensure that our thoughts about God will not be futile, banal, irresponsible, and idolatrous.

Silent, inarticulate religion attempts to escape embodiment and, in doing so, becomes nonsacramental. Particularly in the modern era, the temptation in many quarters has been to relegate God to personal religious experience. "God" becomes a generic symbol, a cipher for realities other than God, a name for "the absence of specific reality, for everything-in-general and nothing-in-particular, for Silence and Nothingness", says Robert Jenson (1978, p. 30).

Perhaps the contemporary call to apophatism is due to some modern thinkers having decided that a negative theology, which stresses a God of the gaps, is the best we can muster in the face of pervasive atheistic secularism. Negation as absence and otherness can seem more realistic, more honest about the fissures in reality, more aligned with the spirit of the age than cataphatic affirmation. Philosopher Gilles Deleuze contends that negation is more exciting than the tedium of mere repetition of historic dogmatic assertions (Deleuze 1994, p. 86). And yet, negation can also be read as capitulation to the pervasive tendency in modern spirituality toward inwardness, privatization, and internalization, the limitation of faith to a personal experience of transcendence, without making bolder claims for God. It is difficult to imagine a Martin Luther King, Jr. or a Dorothy Day arising from the modest claims of negative theology.

## 4. Oppressive Silence

In mysticism and apophatic theology there is a studied lack of attention to the dangers of silence. The silence of those who have nothing to say, the numbed silence of despair after trauma, the eerie quiet after disaster, the muzzling of those who speak truth to power—all these rebuke the unequivocal praise of silence. As Rachel Muers warns, any statement about silence "that affirms it as a phenomenon with positive significance, would be politically and ethically irresponsible. . . particularly. . .in the light of twentieth-century critiques of acts of violent silencing, historical and contemporary" (Muers 2004, p. 10). Silent cowardice, apathy, or indifference, say feminist and womanist writers, must not be lauded. In encouraging pastors to preach against racism, I have found that the most frequent reason given for their not speaking out is, "Fearing that I would say the wrong thing, I said nothing" (Anonymized).

Rachel Muers' theological exploration of silence suggests that a defense of cataphatic theology could be an assertion of male privilege. The silencing of God could be lauded as "the long-awaited silence of the voice of mystifying male authority, that had itself silenced the voices of women, and of countless others who cannot speak from margins" (Muers 2004, p. 46).

And yet, Muers counters that the silencing of God as a defense against "the voice of mystifying male authority" implies that "the speech of God competes with human speech" and fails to see how the "world-constituting character" of God's speech "is the basis of all

creaturely freedom—including the freedom to speak a response to God" (Muers 2004, p. 47). Speech can be oppressive; so can silence. In preaching, the church implies that an Incarnate God compliments, enriches, and encourages rather than competes with human speech.

Muers says, "The association of hearing with the non-activity of humanity before God reinforces a pattern of authority [whereby] the power of the appointed speaker [dominates] those who hear", which is the antithesis of a relational, communitarian, Trinitarian God as revealed by Jesus Christ. Cultural assignation of passive, self-forgetful hearing to women is not integral to the Christian view of God (Muers 2004, p. 112). An aspect of the Pentecostal Good News is that the descent of the Holy Spirit enables the disenfranchised and the silenced publicly to speak: "I will pour out my Spirit upon all flesh, and your sons and your daughters shall prophesy,... upon my servants, men and women, I will pour out my Spirit in those days, and they will prophesy" (Acts 2:17–18).

Despotic, male-enforced silencing of women is not unknown in Scripture: "God isn't a God of disorder but of peace... The women should be quiet during the meeting. They are not allowed to talk. Instead, they need to get under control, just as the Law says. If they want to learn something, they should ask their husbands at home. It is disgraceful for a woman to talk during the meeting" (1 Cor 14:33–35). "I don't allow a wife to teach or to control her husband. Instead, she should be a quiet listener" (1 Tim 2:12).

In 2016 Pope Francis asserted in an ecumenical meeting in Lund, Sweden, that the subject of the ordination of women is silenced from public debate in the Catholic Church forever (Goodstein 2016). Thus Karoline M. Lewis says that the most pervasive challenge that women face in entering ordained Christian ministry is the pressure to be silent and not preach (Lewis 2016).

Cataphatics might note that there is no way for Saint Paul or Pope Francis to be corrected in their attempts to silence women other than through words.

Walter Brueggemann said that a major challenge for any preacher is to "interrupt silence". He recalls Martin Luther King's sermon on 15 April 1967, at Riverside Church in New York, "Beyond Vietnam: a Time to Break Silence". King spoke against U.S. war policy in Vietnam, risking alienation of his followers, even chancing distraction from the perplexing problem of race. Brueggemann says, "'Breaking the silence' is always counter-discourse that tends to arise from the margins of society, a counter to present power arrangements and to dominant modes of social imagination" (Brueggemann 2018, p. 3). The church is dependent upon the silence-breaking speech of preachers who resist the human temptation to leave some things unsaid:

> Alienation and muted rage have a central characteristic in common: an absence of conversation, a loss of speech... Life is reduced to silence. Where there is theological silence, human life withers and dies... In the face of that dread silence, the preacher comes to initiate, to reiterate, to reenact speech that permits the communion for which we so deeply yearn... It is speech and only speech that bonds God and human creatures. The preaching task is to guide people out of the alienated silence of exaggerated self, and out of the silence of denial and rage of an exaggerated God, into a serious, dangerous, subversive, covenantal conversation... Communion is not possible where speech is destroyed... In the midst of these reductions, the preacher is invited to speak in ways that open a world of conversation, communication, and communion. (Brueggemann 1989, p. 49)

## 5. Which God?

Apophatic, negative theology is naïve in its assertions of the virtue of silence and the danger of words. Human experience of silence is not wordless. Words involuntarily invade human consciousness. Silence can be full of meaning only because silence is where language also dwells. Thought follows speech. That which cannot be spoken cannot be thought. The inexpressible cannot be shared, much less examined.



Cataphatic Christians add that utter silence is difficult for humans to achieve because an Incarnate God delights in invading silence with God's self-revelation.

Theologian Eberhard Jüngel sounded like Taylor when he worried that, "God will be talked to death, . . . silenced by the very words that seek to talk about God" (Jüngel 1983, p. vii). But then Jüngel clarified that he is speaking about garrulous, generalized talk about God rather than extolling silence as a theological virtue. Jüngel challenges the theological basis of apophatic theologies, charging that they define God solely on the basis of human limitations, describing God negatively as inexpressible, unknowable, therefore unthinkable, which is the antithesis of Israel's God. To define God's revelation negatively and apophatically, unthinkable and unspeakable, is to negate God, charges Jüngel—mysticism as prelude to atheism. To claim that God is silent or that silence is the only way of humanly approaching God is to appear to be talking about no God or, at the least, a god other than one whom Christians and Jews worship as God.

To say that it is impossible to say anything definitively about God is to begin with the assumption that one knows with certainty the God who one is unable to talk about. One cannot say anything for sure about God except that God is whoever nothing for sure can be said. Begin with a definition of who God is (unsurpassable, incomprehensible, and therefore ineffable) and then see if other divine claimants (like Jesus Christ) fit one's a priori definition of God (Jüngel 1983, p. 236).

For Christians, the temptation to apophaticism is an indication of a failure to work from the implications of Chalcedonian creedal Christianity that designates God as incarnated in Jesus Christ. The formula of Chalcedon affirms that Jesus Christ is fully human, fully divine, God embodied without mixture, confusion, or modification of either the human or the divine.

Amid theological dispute among the followers of Apollinarius of Laodicea (overstressing the incommensurability of the two natures of Christ) and Nestorius (accentuating the distinctiveness and differentiation of Christ's two natures), the Definition of Chalcedon delineated the orthodoxy of Nicaea:

> Our Lord Jesus Christ: the same perfect in divinity and perfect in humanity, the same truly God and truly man . . . like us in all respects except for sin; begotten before the ages from the Father . . . the same for us and for our salvation from Mary, the virgin God-bearer . . . one and the same Christ, Son, Lord, only begotten, acknowledged in two natures which undergo no confusion, no change, no division, no separation; at no point was the difference between the natures taken away through the union, but rather the property of both natures is preserved and comes together into a single person and single subsistent being; he is not parted or divided into two persons, but is one and the same only-begotten Son, God, Word, Lord Jesus Christ, just as the prophets taught from the beginning about him, and as the Lord Jesus Christ himself instructed us, and as the creed of the fathers handed it down to us. (Tanner 2016, pp. 86–87)

The doctrine of Christ's two natures, unified in one person (hypostasis) yet neither confused, mixed, nor separated and detached, identifies Christ as entirely, unreservedly divine and, in the same person, fully, completely human.

From this orthodox formulation of Chalcedon, theologian Karl Barth repeatedly stressed that though humanity may lack the capacity to speak accurately about God, God has taken the initiative to speak to us truthfully of God, to self-represent and self-reveal, in ways that humans hear (Barth 2010, vol. I/1, p. 159). Exodus 3 is at the heart of this divine utterance, in the birth of Israel, when God gives Moses God's name and initiates God-Israel conversation, divine/human dialogue otherwise known as scripture, a dialogue made contemporary and extended by preaching.

God's self-revealing speech precedes all of humanity's talk about God. Human God-talk can only follow as response to God's initiative. God's words can coincide with human words (and vice versa) in unity rather than separation because, in Christ, God has become human. God's words and human words can coexist and inhere in fellowship without merg-

ing or mixing either the divine or the human element. Because God, in the Incarnation, has chosen to be in intimate fellowship with humanity, humans can indeed speak of God without any diminution of God. Jesus Christ reveals God to be the one who has self-determined to be in relationship with humanity and there is no relationship without conversation. God graciously initiates and sustains conversation (and therefore relationship) with humanity. A silent, incommunicative God could never be rationally spoken of as God for us, much less as a God of love.

In the absence of a Chalcedonian, incarnational theology of God's speaking, metaphysical speculation becomes our only way to talk God. On the other hand, in being met by God as Jesus Christ, we learn that God has refused to be relegated to transcendence, rendered vague by mysticism, or trapped by humanity's presumption of oppositional distance between God and humanity. God has positively self-revealed as a Jew from Nazareth, forever breaking silence between us and God.

Incarnation necessitates, encourages anthropomorphism. The Incarnation, as articulated in the definition of Chalcedon, poses the most serious challenge for any who would extoll silence as theologically virtuous (Boesel and Keller 2010). God's sovereignty is not impugned by God's speech to us any more than a lover is less a lover because a lover speaks of love to the beloved. Love insists on expressing itself in speech, declaring, confirming, and reiterating. Because God is love, God speaks and we must and can speak about God (Jüngel 1983, p. 298).

In God's speech, as presented in scripture, God self-presents as relational and social, not arcane and secretive. To be sure, God is a mystery to us, but not because God has anything at stake in appearing mysterious. In revelation, God refuses to be confined to the mysterious, enigmatic, furtive, or cagey, though, as Stephen H. Webb has shown, God's intimate, revealing self-presentation to us may be experienced by us as a challenging mystery due to our preconceptions of God (Webb 2011, pp. 69–72).

Jüngel charged that apophasis has resulted in the "dumb silencing" of God, exclusion of God from contemporary intellectual discourse by our "garrulous silencing" of God. Undue humility, sentimentality, and the substitution of other gods permit the word "God" to continue being used, but in a way that excludes God as speaker. God is rendered mute by the philosophical claim that God is unthinkable and therefore unspeakable, depersonalizing God as the unnamable distant one who is omnipotent except in God's inability to self-identify through human speech (Jüngel 1983, p. 251).

A number of Christian philosophers have charged that apophatic, negative theology is an unwarranted submission to the confines and dictates of contemporary secular philosophy. For instance, Nicholas Lash notes that during the seventeenth and eighteenth centuries, in English and German culture, the word "God" came to name the ultimate explanatory principle rather than the human/divine God of Chalcedonian, orthodox, incarnational Christianity. God was presented as a master lawyer, a mechanic, or technician rather than the One with whom we are in a speaking/hearing relationship. This "god" is the antithesis of the God/human, Jesus, and, while constructed to defend Christianity from secular philosophical assaults, turned out to be a "god" more easily defeated by modern philosophy (Lash 2004, p. 13).

John Milbank opens his Theology and Social Theory by mocking the "false humility" of Enlightenment-tamed Christianity that is hesitant to make strong claims about God, ignoring the testimony of scripture and translating its assertions into more "rational" and "intellectual" ways of thinking (Milbank 1990, p. 1). James Kay says that there is so little focus on preaching as God's speaking in contemporary homiletics because in recent decades, homiletic thought has focused on rhetorical questions ("how" can we speak?) rather than theological designations ("about whom" we are speaking?) because of academic homileticians' attempts to gain "academic respectability" (Kay 2010).

When he charged that preachers have "talked God to death", Jüngel was attacking meaningless, speculative metaphysical talk about God rather than urging apophatic Deism. God constitutes the world and us by speaking, said Jüngel. Theology then "speaks after"

God's primary address. God is no more diminished or degraded through human preaching "than is a lover deprived of . . . power through [the lover's] self-communicating love" (Jüngel 1983, p. 298). It is one thing to say that God is difficult to speak about because God is dissimilar from God's creation. It is quite another thing to disallow God from communion with God's creation.

Theologian Karl Barth dismissed the apophatic tradition as "instructed ignorance", a theological mistake that rests on a failure to recognize that the God of Israel and the church is a speaking subject. Barth repeatedly stressed that it is obvious that human talk about God suffers from human linguistic inadequacy. Yet it is dangerous to say that God's talk to us about God is inadequate (Barth 2010, vol. IV/1, p. 51). While a preference for negative statements about God (apophatic) over positive statements (cataphatic) is often presented as a sign of respect for the unsurpassable greatness of God—God's superiority necessitates God's ineffability—Barth says that apophatics are simply talking about a god who is other than the God we encounter as Jesus Christ. Rendering God ineffable is yet another attempt to refuse the peculiar self-presentation of the Trinity (Anonymized, pp. 76–80).

To assert that "God was reconciling the world to himself through Christ, by not counting people's sins against them. He has trusted us with this message of reconciliation" (2 Cor 5:19) is to identify God as busy reconciling the world through God's speech. There is no means of reconciliation other than through words. It is also to claim that God has elected to do so by entrusting to human agents "this message of reconciliation." God is not just the subject of revelation but its agent whose agency is worked through preachers. Because God has incarnated, human talk about God can hope to be more than simply human projection of human ideas about the divine; human talk can be enlisted by God in God's effort to self-disclose and reveal to God's world.

There are indeed good reasons why it is difficult for humanity to speak of God. However, God is free to speak to humanity of God. That is why, early in his Confessions, Augustine says that he longs to praise God but lacks the words to do so. "Have mercy so that I may find words", prays Augustine (1992, p. 5). Augustine beseeches God for words rightly to speak of and to God. One can only speak about, much less to God, if God verbally self-discloses. All talk about God, truthful talk, is prayer.

## 6. A Silent God as No God?

Kornelis Miskotte contended that scripture, Old Testament and New, is anti-pagan, even though paganism (defined by Miskotte as the creation of multiple deities) is the natural condition of humanity. Creation has an undeniable beauty and humans are tempted to worship nature in its various guises as we project meaning upon the world. The revelation of God in scripture interrupts and judges this temptation by giving us God's own name (Exodus 3). "Paganism projects divine names out of [humans'] experience of life in the world", Miskotte writes. "But humans cannot seize the knowledge of God. It must be given to them" (Miskotte 2021, p. 123; emphasis original). Scripture seeks people who are "bending under the Teaching", that is, reading sacred texts so as to work with the God who has determined to work with people.

Janet Martin Soskice charges that the inexpressible God sounds suspiciously like Feuerbach's "God" as a mere human projection, a construction, a "man-made ideal" (Soskice 2002). A humanly constructed silent God is easier to manage and contain than the God who speaks. Soskice notes that philosopher David Hume implied that apophatism is anthropomorphism in another guise, an attempt to salvage something of "God" without fully answering modern critiques of the notion of God (Soskice 2002, p. 66). In this century, philosopher Anthony Flew took a similar tack, dismissing negative theology as intellectual evasion, killing God by linguistic qualifications. Rather than talk about what cannot be said about God, why not go ahead and be an atheist, asked Flew (Davies and Turner 2002, p. 13). Flew was suspicious that contemporary thinkers speak positively of God only when their concepts of "God" align with their moral and social aspirations, refusing to talk about alleged attributes of God that may challenge the alleged existence of God.

### 7. Because God Speaks, Preachers Can Speak of God

Dietrich Bonhoeffer opens his Christology with the striking statement, "Teaching about Christ begins in silence... This has nothing to do with mystical silence which, in its absence of words, is, nevertheless, the soul secretly chattering away to itself. The church's silence is silence before the Word. In proclaiming the Word, the church must fall silent before the inexpressible... [But] it must be spoken, it is the great battle cry of the church (Luther)... The proclamation of Christ is the church speaking from a proper silence... We must study Christology in the humble silence of the worshipping community". This "humble silence" has nothing to do with the hesychastic "silence of the mystics, who in their dumbness chatter away secretly in their soul by themselves". The silence of the church is "silence before the word", the silent waiting of the church in eager anticipation of God's speaking, the silence of focused listening that is required for any good conversation. This is the silence of listening, listening for a word from God that is not self-derived so that we may speak a word from God (Bonhoeffer 1971a, p. 27). Thus, Bonhoeffer urges fellow Christians to turn away from abstract, speculative assertions about God to a bodily, conversational "encounter with Jesus Christ" incarnate, "God in human form" who is encountered in a rhythm of listening and speaking, address and response (Bonhoeffer 1971b, p. 558).

Barth also spoke positively of church-formed silence when he said that talk about God must occur dialectically, must show "silence as well as speak; it must conceal as well as define; it must negate as well as affirm; it must draw back as well as venture forward" (Barth 1991, pp. 350–60). This is the silence that occurs, for instance, when a preacher studies a biblical text as preparation for preaching, disciplined silence that attempts to restrain one's prejudices and preconceptions in the interest of hearing something fresh and challenging from the text.

It is wise for there to be humility in our talk about God, perhaps especially in our talk about God from the pulpit, but it betrays the Incarnate God to claim that nothing can be said about God except what cannot be said. We humans are severely limited in what we can say about God, and what we say is prone to idolatry. Yet the limitations of our humanity, according to the doctrine of the Incarnation, do not stump God.

Preachers speak of God as those who have been addressed by God. We speak because—to our surprise, delight, and sometimes consternation—God has spoken. God is not arcane and evasive. Our speech about God should be tested, disputed, and contested, but it should not be silenced, indeed cannot be silenced if God is who Christians and Jews claim God to be. As Nicholas Lash says, when it comes to speech about God, "the best remedy for our linguistic insufficiency is to confess it" and then to continue to talk (Lash 2004, p. 13).

Preachers know the near-weekly experience of having to say what is difficult to say and, after saying it, to admit that we could have said it more clearly, more faithfully. While preaching can be an agonistic way of life that never quite achieves its aim—shouting rather than whispering, daring to say definitive statements about God in spite of the pressure to keep silent, combined with intrepid confession of the limits of our speaking—preaching is commanded by Christ and utilized by Christ in his nonviolent salvation of the world. Listening and speaking, otherwise known as preaching, is the thoroughly human way of accessing the truth that has been provided by a God who refuses to allow human limitations to hinder the God/human relationship. As John Henry Newman said, the way to talk about God is by repeatedly "saying and unsaying", though never not saying (Lash 2004, p. 17).

We must exclude the notion "that God only permits one to say what [God] is not" (Jüngel 1983, p. 233). Defining God only by negation is self-contradictory to the preacher Jesus. We cannot be justified, redeemed, or saved by that which we cannot speak. We recognize God and can truthfully speak of God because God permits, even encourages us, to do so. A statement such as "God is love" (1 John 4:8) is meaningless if the lover's love goes unexpressed to the beloved.

Apophatic accounts of God tend not to be about the One who called Israel out of Egypt and raised the crucified Christ from the dead. Divine mysteriousness is not mystery

due to a failure to disclose, but rather is the mystery of how God could be the One who determines to be so close to humanity (Morse 2009, p. 86).

Speaking is a bodily, physical activity. Speech is a public, social, relational endeavor. Silence, on the other hand, is asocial, and solitary, quite the reverse of the Trinity. "Mysticism" would be Barth's name for contemporary spirituality that flirts with apophaticism, the latest attempt to make the church invisible, incorporeal, and unrecognizable as Body of Christ. Mysticism is the way of evasion that contributes little to preaching.

Theologian Robert Jenson joins Barth in celebrating the Logocentric nature of the Christian faith that claims that God is "wholly in Jesus his Word; there is nothing of him left over to be otherwise experienced" through mystical, silent seeking (Jenson 1978, p. 31).

Jenson wonders if nonsacramental silence, rituals, and liturgical actions apart from words, rather than being signs of humility, are other forms of human wish fulfillment and projection, that is, idolatry (Jenson 1978, p. 31). Our tendency toward idolatry, fashioning gods to suit ourselves, gods less demanding than the Trinity, infects human silence no less than human discourse.

Belief in God is belief in divine self-communication. God's reality is relational, communicative, and is otherwise known as Trinity. "It takes two to gospel", claims Jenson, "one to speak and one to hear and turn-about" (Jenson 1978, p. 39). God is available to us solely by divine address, never by our command for divine human communication. God remains God for us not by silencing or hiding, but rather by self-revealing. Apophatics are concerned to preserve the divine difference and distance from humanity by not naming and articulating God. Yet cataphatics could respond that true divine/human distance and difference is based upon what we hear God say, not what we refuse to say about God. The gospel is good news that we cannot, on our own, drift toward or say to ourselves. It must be received from the hands of another, spoken to us. Who among us would say to ourselves or our neighbor, "All who want to come after me must say no to themselves, take up their cross daily, and follow me" (Luke 9:23)?

If Jesus had never preached, we would never have wanted to kill him.

Amid Revelation's cacophony of speeches, prayers, prophecies, and music, "When the Lamb opened the seventh seal, there was silence in heaven for about half an hour" (Rev 8:1). The silence of awe, fear, or wonder? Who knows? Silence is revealing only with words.

## 8. Go, Tell

"Surely you are a god who hides" (Isa 45:15), though not quoted anywhere in the works of Pseudo-Dionysus, has always been a beloved proof text of apophatics. While on occasion Jesus was "oppressed and tormented, but didn't open his mouth" (Isa 53:7), no one took offense at Jesus' prayers or moments of solitude.

Jesus rarely orders people to be silent and, when he does do so, the command to silence is ambiguous. For instance, Jesus once said to his disciples, "Come by yourselves to a secluded place" of silence (Mark 6:31). But the first thing Jesus does after the crowds find him in retreat is to have "compassion" for the crowds. Out of his compassion, Jesus begins to talk to them (6:34).

Sometimes Jesus tells those whom he heals to tell no one (e.g., Matt 8:4//Mark 1:44//Luke 5:14), but he does not say why. He almost never orders his disciples to be silent, except at the Transfiguration (Matt 17:9//Mark 9:9; cf. Luke 9:36). Notably, Jesus orders the storms, the winds, and the demons to keep quiet. We may speculate upon the reasons for Jesus' rare counsels to silence, but we do so only through words.

The Gospel of Mark ends with the women at the tomb commanded by a figure "dressed in a white robe" to "go, tell". The women are silent, "because they were afraid" (Mark 16:8). Eventually the women must have found the courage to speak, otherwise we would not have their testimony. When Jesus is told by the authorities to keep his disciples quiet, he replies, "I tell you, if they were silent, the stones would shout" (Luke 19:40). No devotee of silence was Jesus.

Christian preachers do not enjoy the luxury of being intellectually coy about God. Humble, self-effacing silence is not an option for those who must speak on a weekly basis about the self-revealing, incarnate God who, out of compassion, refuses not to speak. An apophatic God of whom nothing can be said for sure, who evades public discussion and human connection, is not the God who commands and enables preachers to preach. Thus, on a weekly basis, we preachers must overcome our fears and resist the apophatic lure of allowing certain things not to be said, stand up and overtly risk our sermons being used by God in God's self-declared determination to retake God's world.

**Funding:** This research received no external funding.

**Institutional Review Board Statement:** This research did not require institutional review.

**Informed Consent Statement:** This research did not require informed consent.

**Data Availability Statement:** There is no additional data available for this research.

**Conflicts of Interest:** The author declares no conflicts of interest.

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
