# Peer review of "Preaching as Protest against the Apophatic Silencing of God’s People"

_religions, doi:10.3390/rel15020233_

Round 1

Reviewer 1 Report

Comments and Suggestions for Authors

The only comment I have on the text is that the author could gently soften his strongly critical approach to apophatic theology. His objections are valid. However, they do not affect the fact that theology and preaching need a balanced approach to talking about God. Just as one can exaggerate the apophatic approach, so too - by criticizing it too much, without showing its importance and place - one can too easily lose the meaning of cataphatic theology. In general, I think that moving the topic into the space of preaching is a very interesting perspective. However, one must not forget that preaching the Word demands a dose of apophatic awareness. This aspect should be better expressed in the text. 

Author Response

I think that the reviewer makes a good point. In a couple of places I've tried to soften my argument so that it doesn't sound so polemical. I'm pleased that the reviewer was generally pleased with my article and its arguments and seems mainly concerned about the tone, which is a valid concern

Reviewer 2 Report

Comments and Suggestions for Authors

I thought the discussion of apophatic silence was thoughtful and engaged well. However, I found those identified as contemporary apophatics (i.e., Armstrong, Franke, Allison) a bit heavy-handed -- especially the reference to Taylor. Is she an apophatic, or is she merely warning about apophatic speech? This section is not entirely clear to me. Additionally, I am not sure how much value section 6 adds to the overall project. That being said, I found the more homiletic and theological analyses of Barth, Jungel and Bonhoeffer to demonstrate excellent scholarship. The closing challenge ends the essay on a strong note.

Author Response

I'm grateful for this reviewer's positive feedback on the article. I did highlight B.B. Taylor and her comments because they were here Beecher Lectures at Yale and she herself published a book in which she praised the virtue of keeping silence about God. While she may not be formally an "apophatic," she is certainly, by this book, an advocate of apophatism.

While Alison's book praising apophatic theology seems a bit out of sync with the rest of his work as a New Testament scholar, it was certainly a theme during an important period of his work. therefore it seems fair to engage him on this theme.

I'm delighted that the reviewer approved of my attempt to engage apophatic theology from a homiletic point of view and that the reviewer liked my ending. Thanks.

Reviewer 3 Report

Comments and Suggestions for Authors

Consider the implications for persons who are nonverbal.  Is there communication with God or God talk that includes nonverbal communicators?

Lines 221-226. What is the reader to do with these lines?  How do they function in this writing?  As they are, the reader assumes they will be addressed or explained, but they are not.

Line 426-432.  Consider revising redundancy.

Author Response

I'm pleased that the reviewer generally found my article to be helpful and of positive value. I thought the reviewer made a good point about nonverbal persons. I considered treating that subject but thought that it could take me too far afield in an already rather long article. I was reluctant to engage the question because I thought to treat it fairly would be a big undertaking.

As for the comments about lines 221-226, these are scripture quotes that mainly serve as examples of the silencing of women in places in scripture. I thought they therefore serve the reader as a reminder of unjust silencing of women and didn't deserve or require a detailed explication of their obviously derogatory comments about the public speaking of women in church. They are rather jarring and I didn't want to soften that jarring quality.

I can certainly see the reviewer's point about "redundancy" at numbers 426-432. However, I see these sentences as a sort of summation, toward the end, of my rather wide ranging arguments in the first two thirds of the article. While I can understand that they might sound redundant to some readers, I hope they will be a helpful summation and reiteration for others.